# Continuous Production of Highly Tuned Silk/Calcium-Based Composites: Exploring New Pathways for Skin Regeneration

**DOI:** 10.3390/molecules27072249

**Published:** 2022-03-30

**Authors:** Anabela Veiga, Rui Magalhães, Marta M. Duarte, Juliana R. Dias, Nuno M. Alves, Ana Rita Costa-Pinto, Filipa Castro, Fernando Rocha, Ana L. Oliveira

**Affiliations:** 1CBQF-Centro de Biotecnologia e Química Fina—Laboratório Associado, Escola Superior de Biotecnologia, Universidade Católica Portuguesa, Rua Diogo Botelho 1327, 4169-005 Porto, Portugal; s-anveiga@ucp.pt (A.V.); rsmagalhaes@ucp.pt (R.M.); s-msmduarte@ucp.pt (M.M.D.); aloliveira@ucp.pt (A.L.O.); 2Laboratory for Process Engineering, Environment, Biotechnology & Energy, Department of Chemical Engineering, Faculty of Engineering, University of Porto, Rua Dr. Roberto Frias, 4200-465 Porto, Portugal; 3ALiCE-Associate Laboratory in Chemical Engineering, Faculty of Engineering, University of Porto, Rua Dr. Roberto Frias, 4200-465 Porto, Portugal; juliana.dias@ipleiria.pt; 4Centre for Rapid and Sustainable Product Development, Instituto Politécnico de Leiria, 2430-028 Marinha Grande, Portugal; nuno.alves@ipleiria.pt; 5i3S-Instituto de Investigação e Inovação em Saúde, Universidade do Porto, Rua Alfredo Allen 208, 4200-135 Porto, Portugal; anap@i3s.up.pt; 6IPATIMUP-Instituto de Patologia e Imunologia Molecular da Universidade do Porto, Rua Júlio Amaral de Carvalho 45, 4200-135 Porto, Portugal

**Keywords:** calcium phosphate-based materials (CaP), cerium (Ce), human dermal fibroblasts (HDFs), modular oscillatory flow plate reactor (MOPR), silk sericin (SS), skin regeneration

## Abstract

Calcium plays an important role in barrier function repair and skin homeostasis. In particular, calcium phosphates (CaPs) are well established materials for biomedical engineering due to their biocompatibility. To generate biomaterials with a more complete set of biological properties, previously discarded silk sericin (SS) has been recovered and used as a template to grow CaPs. Crucial characteristics for skin applications, such as antibacterial activity, can be further enhanced by doping CaPs with cerium (Ce) ions. The effectiveness of cell attachment and growth on the materials highly depends on their morphology, particle size distribution, and chemical composition. These characteristics can be tailored through the application of oscillatory flow technology, which provides precise mixing control of the reaction medium. Thus, in the present work, CaP/SS and CaP/SS/Ce particles were fabricated for the first time using a modular oscillatory flow plate reactor (MOFPR) in a continuous mode. Furthermore, the biological behavior of both these composites and of previously produced pure CaPs was assessed using human dermal fibroblasts (HDFs). It was demonstrated that both CaP based with plate-shaped nanoparticles and CaP-SS-based composites significantly improved cell viability and proliferation over time. The results obtained represent a first step towards the reinvention of CaPs for skin engineering.

## 1. Introduction

Calcium phosphates (CaPs) are well recognized materials for bone tissue engineering (TE). However, new possibilities have arisen for these standard bioceramics. Recent studies have been focused on reinventing CaPs for skin-TE [1]. Calcium (Ca) plays an important role in the barrier function repair and skin homeostasis and serves as a modulator in cell proliferation and differentiation, promoting wound healing [2].

In this context, the combination of CaPs with other molecules or materials to promote skin tissue regeneration becomes an attractive strategy, providing appropriate microenvironments for cell interaction. Recently, we have proposed -nano and–micro CaP/sericin composites as suitable candidates for TE approaches [3]. Silk sericin (SS), until recently considered unfit for biomedical use, is now accepted as a valuable byproduct from the textile industry, able to stimulate collagen production and to have antioxidant, moisturizing, and anti-inflammatory properties [4]. Furthermore, SS is reported to promote the growth of fibroblast cells [5] and to increase cell adhesion and proliferation of mammalian cells [6]. When used as a 3D matrix, this natural protein also provides cell proliferation [7,8]. Several works have reported that adding sericin to CaP precipitation systems can improve cell behavior [4,7,8,9]. In the last decade, researchers have shown that hydroxyapatite (HAp)/sericin particles increase adhesion and proliferation of several cell types such as Human Bone Marrow Derived Mesenchymal Stem Cells (BMSCs) [7], MG63 cell line [9], or MC3T3E1 cells [8].

Another effective approach to enhance the biocompatibility of CaP/SS composites, and consequently, their application for TE, is to add dopants. Cerium (Ce), for example, is being increasingly used due to its antibacterial properties [10]. Chigurapati and co-authors [11] have shown that Ce-based nanoparticles accelerate the healing of dermal wounds by a mechanism that involves enhancement of the proliferation and migration of fibroblasts, keratinocytes, and vascular endothelial cells.

The effectiveness of cell attachment and proliferation on CaP composite particles highly depends on their physicochemical characteristics, which in turn is directly dependent on the processing strategy. Continuous flow regimes operated through oscillatory flow reactors (OFRs) provided with smooth periodic constrictions (SPCs) improved mixing efficiency and mass transfer processes in multiphase systems, generating a product with uniform and controlled characteristics [12,13]. The liquid or multiphase fluid is typically oscillated in the axial direction and this motion interacts with the SPCs forming vortices, the mixing intensity being controlled by the oscillation frequency (*f*) and amplitude (*x*_0_) [12,14]. Moreover, these reactors allow upscaling by adding identical meso-OFRs in series without compromising the chemical composition and efficiency [12,15].

In the present work, we produced CaP-sericin (CaP-SS) and CaP-sericin-cerium (CaP-SS-Ce) particles for skin regeneration, applying a continuous coprecipitation process, using a specifically designed modular oscillatory flow plate reactor (MOPR) (WO/2017/175207).

The physicochemical characteristics of the materials were evaluated, while the in vitro behavior was assessed for the first time using Human Dermal Fibroblasts (HDFs) over 7 days of culture and compared with pure CaP particles previously developed [16]. The selected pure CaP formulations (CaP1-nano, CaP2-nano/micro, and CaP3-micro) present different characteristics (CaP phase, morphology, size, Ca/P molar ratio, crystallinity), and allow for us to study the influence of these properties on cell behavior and to compare them with the newly produced composite formulations.

All experimental conditions that correspond to nano and low crystalline HAp-based particles (CaP1-nano, CaP2-nano/micro, and CaP-SS) considerably promote HDFs adhesion and viability, contrary to what was observed for CaP3-micro particles, which correspond to micro-scale and high crystallinity brushite. On the other hand, the addition of SS significantly increases the amount of DNA produced. A similar trend was verified for CaP2-nano/micro, in which the seeded cells also presented a highly organized cytoskeleton structure.

This work represents a first step towards the reinvention of CaPs for skin TE, demonstrating the importance of physicochemical and compositional characteristics of CaPs in the biological response of skin cells.

## 2. Results

### 2.1. Physicochemical Properties of Produced Calcium-Based Particles 

#### 2.1.1. Phase Identification 

Functional groups of HAp were confirmed by FTIR analysis (Table 1) for CaP1-nano, CaP2-nano/micro, CaP-SS, and CaP-SS-Ce. Most bands attributed to the phosphate group, PO_4_^3−^, are exhibited [17]. The vibrational mode of OH^−^ (630 cm^−1^) was not identified in the synthetized samples [16]. 

Brushite characteristics peaks were identified for CaP3-micro. Bands at around 541 cm^−1^ and 574 cm^−1^ are attributed to the ν4 bending vibrations of the P-O-P mode [18]. The peaks identified in Table 1 between 3533 and 3151 cm^−1^, result from νO–H vibration mode of H_2_O. The H-O-H bending gives rise to absorption at 1647 cm^−1^. The absorptions at 1205 and 1099 cm^−1^ are due to P=O associated stretching vibrations. The P-O-P asymmetric stretching vibrations give rise to absorptions at 777 and 655 cm^−1^ [19,20]. Concerning CaP-SS and CaP-SS-Ce experimental conditions, amide functional groups were identified. 

The XRD analysis of the prepared powders (Table 2) revealed no secondary phases besides HAp for conditions attributed to CaP1-nano and CaP-SS. In these samples, the maximum peak intensities match the °2 Theta values attributed to HAp reference pattern (JCPDS 9-0432) and to commercial HAp (Merck, Darmstadt, Germany, Hydroxyapatite powder). On the other hand, CaP3-micro pattern is similar to brushite reference (JCPDS 72-0713) and CaP2-nano/micro pattern exhibits typical peaks of both HAp and brushite. Further CaP3-micro pattern has sharp peaks with high intensity, while CaP1-nano and CaP-SS patterns have broad peaks with lower intensity. A combination of these peaks is evidenced in the CaP2-nano/micro pattern. For CaP-SS-Ce, no peaks were identified (Figure 1).

Ca/P determination was performed to the fabricated samples, confirming the formation of CaP. Regarding CaP-SS-Ce, EDX allowed the identification of cerium. The measured Ca/P molar ratio of CaP1-nano, CaP2-nano/micro, and CaP-SS is similar to the stoichiometric value of HAp (1.67) and lower for CaP3-micro and CaP-SS-Ce (≈1) (Table 3).

#### 2.1.2. Size, Morphology, and Crystallinity

The obtained particles have different morphologies, crystallinity, and size, according to the results obtained by TEM (Figure 2), SEM (Figure 3), and laser diffraction (Figure 4), respectively.

Based on SEM images (Figure 3), commercial HAp is composed by microspheres with different diameters ranging from less than 1 µm to more than 3 µm. On the other hand, from TEM analysis, it is possible to verify that these microstructures are composed by elongated aggregated nanoparticles (*d*_50_ ≈ 80 nm). Rod-like aggregated nanoparticles were obtained in CaP1-nano. On the other hand, for CaP2-nano/micro, in addition to nanoparticles with rod-like shape morphology, it is also possible to visualize micro particles with a plate-like shape. CaP3-micro presents the morphology of large plates and is associated with a higher mean particle size (*d*_50_ ≈ 6 µm). Regarding the experimental conditions with sericin, while CaP-SS corresponds to sharper nanorods (more plated) (d50 = 0.080 μm), CaP-SS-Ce presents micro and nano plates (*d*_50_ ≈ 500 nm).

### 2.2. In Vitro Cell Viability and Cell Behavior 

#### 2.2.1. MTT Assay and DNA Quantification 

The cell viability and proliferation of HDFs (human dermal fibroblasts) when in contact with the CaP and CaP-based composites was investigated by MTT assay and DNA quantification (Figure 5 and Table 4 and Table 5).

According to Figure 5A, the absorbance at 570 nm increased over time for all experimental conditions, the increment being more pronounced between the 3rd and the 7th day of cell culture. Furthermore, the only statistically significant difference was registered on day 7, between CaP3-micro (brushite) and the control (*p* < 0.01). The majority of the particles produced resulted in high cell viability (>80%) (Table 4). Regarding the DNA quantification results, represented in Figure 5B, a similar trend over time was registered with an increasing DNA amount, with the exception of the CaP3-micro sample, which was composed of brushite. After 7 days of culture, DNA content was significantly higher for cells in contact with the silk containing composites, CaP-SS (*p* < 0.0001), CaP-SS-Ce (*p* < 0.0001), and for CaP2-nano/micro (*p* < 0.01, *p* < 0.001) (Table 5).

#### 2.2.2. Cell Morphology

The analysis of the cells seeded and cultured in the several experimental conditions by SEM and confocal microscopy is presented in Figure 6, Figure 7 and Figure 8. 

SEM micrographs in Figure 6 show that after 1 day of cell culture, most cells adopted a rounded and flat morphology. On day 3, there was a considerable increase in cell density for all studied conditions and HDFs became more elongated. After 7 days, a cell monolayer was formed. This effect is particularly clear for CaP2-nano/micro, CaP-SS, and CaP-SS-Ce, presenting high confluency. CaP1-nano particles also led to the formation of a cell monolayer, contrary to commercial HAp and CaP3-micro. In Figure 7, the CaP-SS-Ce particles are represented in Z-contrast mode to evidence the presence of cerium. Since this element has a high atomic number detected in Z-mode, it allows us to clearly demonstrate the presence of the particles which were put in contact with HFDs. Its presence was further confirmed by performing localized EDX (Table 3). It was also found that some particles form aggregates, which is consistent with the particle size distribution by volume (Figure 4B).

Confocal images (Figure 8) show the nuclei and cytoskeleton of the cells, as well as their distribution, alignment, and morphology. After 1 day of cell culture, cell adhesion was visible for all conditions in contact with the produced particles. The HDFs showed spindle and elongated-shape morphology, with actin filaments stained in red by phalloidin. After 7 days, the cells were more scattered and exhibit several contact points among themselves. 

The samples of commercial HAp, CaP1-nano, and CaP2-nano/micro exhibit a very symmetrical and oriented cytoskeleton. In addition, the HAp samples (CaP1-nano, CaP2-nano/micro, CaP-SS, and CaP-SS-Ce) produced in the MPOFR appear to have more cells when compared to commercially available HAp. This is particularly visible in CaP2-nano/micro particles, in which the confluence degree and number of oval nuclei seem higher, as well as a more organized cytoskeleton structure. In the composite particles with sericin, the CaP-SS evidence cells with a well-organized cytoskeleton, while the CaP-SS-Ce present HDFs that are more heterogeneous. A less organized cytoskeleton was also evidenced for CaP3-micro experimental conditions, which also appears to be associated with the presence of a small number of cells.

Regarding the antimicrobial activity of CaP-SS-Ce samples, it was found that while the growth of Gram– was inhibited, the same was not true for Gram+ bacteria. Furthermore, no changes were observed with increasing concentrations (Figure 9).

## 3. Discussion

### 3.1. Physicochemical Characterization (FTIR, XRD, EDX, SEM, TEM, Laser Diffraction)

According to FTIR analysis, CaP1-nano, CaP2-nano/micro, CaP-SS, and CaP-SS-Ce have characteristic functional groups of HAp, while CaP3-micro has the characteristic bands attributed to brushite (also known as dibasic calcium phosphate dihydrate). In a previous work [16], we showed that by changing the initial concentration of the precursor solutions, different CaP phases such as brushite could be obtained. In HAp-based particles, the absence of the OH^−^ stretching mode peak at 3572 cm^−1^ can be the result of overlapping with the adsorbed water band (3700 to 3000 cm^−1^), which is commonly present in wet precipitation reaction systems [3,15]. Furthermore, CO_3_^2−^ bands suggest that carbonated HAp is formed. The carbonate group can substitute PO_4_^3−^, giving rise to B-type carbonation [21]. In addition to CaP characteristic peaks, in the experimental conditions with sericin (CaP-SS, CaP-SS-Ce), amide groups were identified. This compound plays a critical role in the structure and function of proteins, linking amino acids (peptide bonds) [22]. Regarding CaP3-micro identified peaks, bands at around 541 cm^−1^ and 574 cm^−1^ can be attributed to the ν4 bending vibrations of the P-O-P mode [18]. The peaks identified in Table 2 between 3533 and 3151 cm^−1^ and 1647–653 cm^−1^, result from νO–H vibration modes and δO-H bend modes of the water in brushite [19,20] (Table 1).

The XRD analysis corroborated the FTIR results for CaP1-nano, CaP-SS, and CaP3-micro. XRD patterns of CaP1-nano and CaP-SS correspond to HAp and the pattern of CaP3-micro agrees with brushite reference. The results suggest the synthetized particles corresponding to HAp and brushite have a hexagonal and monoclinic crystal system, respectively [23,24]. On the other hand, the presence of both HAp and brushite peaks in CaP2-nano/micro particles indicate that both CaPs are present. Experimental conditions associated with CaP1-nano and CaP-SS composite particles have a broad XRD spectra, characteristic of samples with low crystallinity. Moreover, when evaluating the effect of SS on these samples, produced using the same oscillatory flow mixing, it was possible to verify that the crystallinity of particles with SS appears to be even lower, given the XRD peaks with less intensity. 

CaP that is synthetized using wet chemical precipitation and physiological conditions of temperature (37 °C) and pH (6–9) often results in HAp particles with poor crystallinity [3,13,25,26]. CaP-SS-Ce particles have an amorphous structure, since no peaks were identified (Table 2, Figure 1). In the work of Pharai et al. [27], HAp/Cerium particles were studied, and it was found that when cerium ions are incorporated into the HAp lattice, XRD peaks are broadened, which indicates a decrease in crystallinity. This phenomenon can be due to a different charge compensation mechanism during the substitution of calcium by cerium ions [27]. On the other hand, brushite (CaP3-micro) has sharp peaks which indicates a highly crystalline structure [16] (Appendix A).

Through EDX technique and Ca/P molar ratio determination, the composition of the produced particles was assessed. In addition to calcium and phosphate elements, oxygen was also detected in all experimental conditions resulting from the chemical composition of CaPs, which is Ca_10_ (PO_4_)_6_(OH)_2_ for HAp and CaHPO_4_·2H_2_O for brushite [28]. According to the elemental composition, CaP1-nano and CaP2-nano/micro particles have a Ca/P ratio slightly higher than stochiometric HAp (1.67) [26]. This is characteristic of carbonated HAp and in line with FTIR analysis. Exceptions include CaP3-micro and CaP-SS-Ce, which have the lowest Ca/P ratio. CaP3-micro, identified as brushite in both FTIR and XRD, has a Ca/P closer to the molar ratio reported for brushite (Ca/P = 1). Regarding CaP-SS-Ce, the presence of Cerium was identified in EDS, representing approximately 30% (*w*/*w*) of the material, which probably resulted in a decrease in the total amount of Ca and P. However, the reduction of Ca/P ratio is due to the low Ca amount, characteristic of calcium-deficient apatite. In HAp, the substitution of Ca^2+^ by Ce^3+^ can occur in co-precipitation systems [29]. In the work of Vieira et al. [10], calcium-deficient HAp-Ce composites were synthetized and also exhibited a lower Ca/P molar ratio than stochiometric HAp.

Regarding size and morphology, TEM (Figure 2) evidences the size and the morphology of primary CaPs particles. CaP1-nano is composed of aggregated nanoparticles in the form of rods similar to other HAp particles obtained using wet chemical precipitation [3,12,13,30]. Similar nanorods organized in microcapsules are present in commercial HAp. This morphology is characteristic of particles obtained by spray drying [31]. CaP2-nano/micro is composed by particles in the form of aggregated nanorods and microplates, which may be consistent with a HAp/brushite biphasic material, also indicated by the XRD patterns. CaP precipitation is dictated by both thermodynamic solubility product and kinetic factors. Thus, even though HAp is the most stable CaP under the reaction conditions used, other CaP phases can be formed [32]. By changing the mixing conditions (oscillation *x*_0_ and *f*) used in the MOPR and the chemical parameters (initial reagents concentration) of the precipitation reaction, CaPs with different characteristics can be obtained [16,33]. Brushite obtained in CaP3-micro is mainly composed by microplates, also presenting some nanoparticles surrounding the bigger structures. The synthetic crystallization of brushite usually yielded plate and star crystals, which resemble the morphology of crystals present in urinary stones and pathological joints [34].

The CaP-composite particles also exhibit distinct characteristics (Figure 2). In terms of morphology, CaP-SS particles adopt a plate morphology rather than a rod morphology, like those of CaP1-nano. In a previously published work [3], where the influence of SS was assessed, increasing SS concentration was correlated with an increase in the mean particle size and the number of plate-like particles, as well as an increase in the aggregation degree and a decrease of the crystallinity. The plate-shape morphology can be induced by the amino acids of sericin such as Gly, Ser, Glu, and Asp. These are reported to lead to the synthesis of plate-shaped structures [35]. CaP-SS-Ce is characterized by large and mostly smooth plaques, similar to other published works [27]. 

From SEM (Figure 3) analysis, the morphology of aggregates corroborates the TEM observations. While CaP1-nano is composed by nanoparticles, CaP2-nano/micro also presents particles on a micro scale. The well-defined edges identified in CaP3-micro are characteristic of high crystallinity as confirmed by XRD analysis. More elongated and sharp plate-like particles are visible for Ca-SS particles. Moreover, the addition of Ce to Ca-SS resulted in the formation of plaques. The backscattered image of this sample in SEM Z-contrast mode (Figure 7) shows the presence of cerium due to its atomic structure composition.

Considering particle size distribution (Figure 4), the high degree of aggregation is confirmed by size distribution in volume (Figure 4B). The formation of clusters linked by weak cohesion forces (van der Waals) is common in precipitation systems [33]. The tendency to form aggregates results from the high surface tension of nanoparticles [36,37]. This behavior is reported in several works focusing on the precipitation of HAp and HAp-based composites [3,12,15]. In addition, the suspensions were not subjected to any treatment to separate the aggregates before being analyzed by laser diffraction. On the other hand, the particle size distribution confirms that CaP3-micro are the produced materials with larger size, followed by CaP-SS-Ce, which was also evident by SEM and TEM microscopy. Nano-size is attributed to CaP1-nano, CaP2-nano/micro, and CaP-SS. It should be noted, that although CaP2-nano/micro also has microstructures, most of them are most likely nano-sized.

### 3.2. In Vitro Evaluation (SEM, Confocal, MTT, DNA)

Regarding the in vitro behavior of HDFs when in contact with the fabricated particles, in Figure 6 and Figure 8, it is possible to verify that the cells grew over time, involving and covering the synthesized particles. Brushite (CaP3-micro), however, did not promote cell proliferation as compared with the control and all the studied conditions (Table 3 and Table 4). Although brushite is typically considered as a biocompatible and biodegradable material, higher crystallinity is often linked with less efficient cell growth [38]. Brushite can be converted in HAp at a pH > 7, which is characteristic of culture mediums such as DMEM. 

Differences between CaP and CaP-SS-based particles were shown in confocal images after 7 days of cell culture (Figure 8). While for the cells in contact with HAp particles (commercial HAp, CaP1-nano, and CaP2-nano/micro), the cytoskeleton is more oriented and a higher number of nuclei is visible, for the cells seeded with CaP-SS particles, in particular CaP-SS-Ce, the cytoskeleton is less organized, and the nuclei are fewer and larger. Thus, the addition of cerium seems to diminish cell orientation. 

The mechanisms that govern cellular orientation responses are still scarce. However, the substrate on which cells are cultured is a contributing factor [39]. In the work of Gliga et al., cerium oxide particles interfere with cytoskeletal organization of neural C17.2 murine cell line, showed by a reduction of neuron-specific β3-tubulin expression and a neuroglial marker (GFAP-glial fibrillary acidic protein) [40]. The physicochemical properties of the CaP-SS-Ce particles, namely size, can also have implications in cellular organization. Unlike the other nano-HAp-based samples, particles loaded with Ce are in the micro scale. In addition, the particle size distribution in number (Figure 4A) is broader, which may indicate the presence of particles of distinct sizes, resulting in implications in cellular organization.

Regarding the HAp samples, CaP2-nano/micro promoted high cell alignment and accelerated migration, exhibiting enhanced cell proliferation. Fibroblast adhesion promoted the assembly of actin cytoskeleton, which generated mechanical forces needed for structural support [41]. In comparison, cells in the presence of CaP-SS have a slightly lower degree of organization, being similar to those in contact with commercial HAp.

The presence of synthetized particles was evidenced in the CaP-SS-Ce condition using Z-contrast (Figure 7). It would be expected that brushite particles (CaP3-micro) would be more easily identified in SEM images due to their size. However, as mentioned earlier, brushite may have been converted to HAp when in contact with the culture medium [42].

The evaluation of cell behavior and in vitro cell viability assessed by MTT demonstrated that, according to the ISO standard ISO10993-5, most of the materials had viability levels compatible with no cytotoxic effects, i.e., above 70%, when compared to the negative control (Table 3) (Equation (1)). 

Statistical analysis revealed that the only significant differences occurred after 7 days, between CaP3-micro and the control (*p* ≤ 0.01). In fact, cell viability in brushite particles was approximately 67.71% (Table 3), which was the lowest cell viability in line with the SEM images (Figure 6). Brushite has high solubility in physiological conditions, which can affect intracellular homeostasis. On the other hand, HAp particles are more stable in these conditions [33].

In physiological conditions, when brushite is converted to HAp, an inflammatory response is observed due to the large amounts of released acid during the reaction [42]. Hence, brushite is not conventionally used to promote cell adhesion and proliferation, being more used as a cement paste [43] or as an intermediate for tooth mineralization [42,44]. Regarding the effect of crystallinity, according to Seydlová et al. [45], this physicochemical property of CaPs does not affect the level of keratinocyte differentiation in coating materials used in dental implantology.

Concerning DNA quantification, DNA increased between 1 and 3 days of cell culture. However, no significant differences between the experimental conditions were identified. Additionally, a pronounced increase in the DNA amount was verified at day 7. CaP2-nano/micro, CaP-SS, and CaP-SS-Ce were associated with a higher DNA amount (2.20 ± 0.21, 2.88 ± 0.15, and 2.94 ± 0.49 ng/mL, respectively).

Accordingly, the addition of sericin leads to a significant increase in DNA quantification (CaP-SS, CaP-SS-Ce) (*p* ≤ 0.0001). The effect of this silk protein is also noticeable in SEM and confocal images after 7 days of cell culture, where cells are confluent (Figure 6 and Figure 8). These results are in agreement with the reported studies on the effects of CaP/sericin composites, in which an enhanced cell behavior is generated [4,5,7,8,9].

Although the effect of sericin on CaP synthesis is still not well clarified, in a previous work, we proposed a mechanism for the assembly the CaP/sericin nanocomposites, in which the SS amino acids play a crucial role in the process, establishing strong interactions with Ca^2+^ and PO_4_^3−^ ions. This can cause a local supersaturation that forms nuclei, which can develop larger plate-like HAp particles, thus playing a critical role in HAp nucleation and growth [3].

Liu and colleagues [46] stated that cells cultured with sericin show upregulation of genes associated with cell energy metabolism and DNA replication, such as cellular respiration (*HMGCS1*, *TXNIP*, and *ABCA13*) and cell cycle progression (*CKS1*, *RMRP*, *FN1*, and *MFSD8*).

The findings of our in vitro tests may also mean that the presence of plate-shaped nanoparticles, whose formation is favored by the addition of sericin [13], can influence DNA production. Interestingly, the precipitated HAp presenting plaque-shaped particles (CaP2-nano/micro) has also a higher DNA amount (*p* ≤ 0.01–0.001), which can indicate that this type of morphology positively affects cell behavior. However, Xu and co-authors [47] studied the effect of different HAp morphologies on cytotoxicity evaluation (MTT, ALP activity, and apoptosis) and concluded that cell behavior depends on both particle’s concentration and specific surface area, rather than its shape. Other studies defend that biphasic CaP systems, such as HAp/brushite, lead to the presence of a micro-scale texture combined with a surface roughness that is suitable for cell anchorage and development. Additionally, the conglomerate isles can act as center points for first cell-material interaction [48,49].

Overall, changing the oscillatory flow mixing conditions can affect the resulting biological properties of the synthetized particles. In this case, by increasing *f* from 1.9 Hz to 4 Hz, it is possible to obtain particles that stimulate cell proliferation (Figure 2 and Figure 3 and Table 4). The DNA amount for rod-like particles produced in the MOPR (CaP2-nano/micro) is similar to commercial HAp and to the report by Castro et al. [12]. 

### 3.3. Antibacterial Activity

The results obtained indicate that produced particles can induce an inhibitory effect on *Escherichia coli*, which can cause severe infections in skin or soft tissues and therefore can be useful for tissue regeneration application [50]. According to the results of in vitro assays, the number of incorporated particles is non-toxic and confers antibacterial properties. However, a more complex study on the antibacterial properties of cerium is required, namely by the effect of used concentrations of HAp, sericin, and cerium. 

In addition to allowing the visualization of the synthesized particles and its distribution in vitro, cerium is a very interesting and undervalued material for skin-related applications. When combined in low concentrations with HAp, Ce^3+^ ions confer antibacterial properties [51]. Several studies reported that Ce^3+^-substituted HAp has excellent microbial restriction against *S. aureus*, *E. coli*, and *Lactobacillus pathogens* [52]. According to a study conducted by Ciobanu and co-authors [53], the antibacterial properties induced by HAp/Cerium against *Escherichia coli* and *Staphylococcus aureus* bacteria increase with increasing cerium content, being more effective against *E. coli*.

### 3.4. Potential Application in Skin-TE

The manufactured particles have great potential for the area of skin-TE. The reported studies on CaP/sericin particles usually focus on bone-TE and drug delivery systems [4]. However, the presented study opens new avenues for these composite materials for skin TE. In fact, to the best of our knowledge, only a few studies have proposed the use of calcium-based nanoparticles for skin wound healing [1,54].

The materials herein produced can be safely used for direct application in skin wounds. In clinical practice, the direct topical application of powders for wound healing is a common procedure and usually employs biomolecules such as collagen, elastin, or fibronectin [55,56]. Ca-based materials modulate calcium homeostasis, and as this is a key messenger involved in several signaling cascades critical to wound healing, are an added value [54]. In a study carried out by Kawai et al. [57], calcium-based nanoparticles decreased open wound size via contracture mediated by the release of ionized calcium into the wound bed. When combined with sericin, the biological properties of calcium-based particles can be further improved as demonstrated in the present study. Sericin powders are commercially available for skin care, especially in the Asian market, where sericulture is traditionally widespread [58]. These powders provide humidity control and antioxidant properties [59,60], useful not only for cosmetic purposes but also for skin repair and regeneration. Furthermore, silk sericin-based materials promote the attachment, viability, growth, and differentiation of a variety of cell types, modulate the release of bioactive molecules and drugs, and influence the innate immune response of the host [61]. Other applications have been reported for silk-sericin based materials. Liu and co-authors stated that sericin-supplemented medium perform as well as, or even better than, those cultured in FBS-containing medium [46].

In addition to the immediate use of the sympathetic particles, 3D-based systems with CaP/sericin particles fabricated with the MOFPR technology provide standardized conditions to develop reproducible cell-based materials for skin TE or to establish physiologically relevant in vitro models to test pharmacological agents. 

## 4. Materials and Methods

### 4.1. Description of the Modular Oscillatory Flow Plate Reactor

The precipitation of CaPs and CaP composites was carried out in a continuous mode using a modular oscillatory flow plate reactor (MOFPR) (75 mL). The experimental apparatus was composed of a jacked plate-like reactor provided with 2D-SPCs presented in two parallel faces of the rectangular cross section tube, a custom-built mixing chamber, and a linear motor (Festo, EMME-AS-100-S-HS-AM) to oscillate the multiphase fluid. The set-up and the dimensions of each constriction are represented in Figure 10.

The reagents were continuously injected into the MOFPR using a peristaltic pump (BT300-2J, Longer Pump, Hebei, China). The temperature of the thermostatic bath connected to the MOFPR (Tectron-Bio, JPSELECTA, Rhode Island, EUA) was adjusted to allow a temperature at the outlet of the reactor of 37 °C. Both temperature and pH of the final suspensions were monitored with a thermocouple and pH electrode (SenTix Mic-D, WTW, Porto Salvo, Portugal).

### 4.2. Particles Synthesis 

CaP particles were obtained by mixing equal volumes (500 mL) of a solution of CaCl_2_.2H_2_O (Merck, 99.5%) and a solution of Na_2_HPO_4_ (Sigma-Aldrich, St. Louis, MI, USA 99.0%). The initial Ca/P (calcium/phosphate) molar ratio was 1.67, characteristic of stoichiometric HAp. CaP composites were obtained by dissolving sericin or sericin/CeCl_3_.7H_2_O (Sigma-Aldrich, 99.0%) in the calcium precursor solution. The synthesis route implemented maintains near-physiological conditions of temperature (37 °C) and pH (9–6). pH monitoring was achieved by using a pH electrode (Sentek, P11/HA). The pH decreases until it stabilizes at around 6 for HAp particles and around 5 for brushite, which is indicative of the formation of the most stable CaP phase under the reaction conditions used [3].

Sericin solution was obtained by a previously reported simple extraction in boiling water method [3]. Briefly, cocoons from *Bombyx mori* silkworm (Silk Museum of Portuguese Association of Parents and Friends of the Mentally Deficient Citizen of Castelo Branco) were cut and cleaned into small pieces (~1 cm) and added to boiling ultrapure water (18.3 MΩ·cm^−1^ at 25 °C) in a proportion of 1:100 (*w*/*v*) for 60 min. According to previous work by Baptista Silva et al., the extraction method used results in a molecular weight distribution between 40 and 440 kDa, with the highest expression between 100 and 200 kDa [63]. The experimental parameters used were based on preceding work, in which the influence of the oscillation amplitude (*x*_0_) and frequency (*f*), residence time (τ), initial reagent concentration, and other physicochemical parameters were investigated on the final CaP physicochemical properties [16]. The particles with the most promising composition, morphology, mean size, and particle size distribution were reproduced for the current paper, in order to evaluate the effect of these parameters on their biological behavior and to compare with the developed calcium-sericin (CaP-SS) and calcium-sericin/cerium-based composites (CaP-SS-Ce) (Table 6). CaP-SS concentration was based on the literature, and results on a final composition of 80/20 wt.% HAp/SS [3].

### 4.3. Particles Characterization

Suspensions were withdrawn at the outlet of the MOFPR, filtered (0.2 µm pore size membrane, Gelman Sciences, Ann Arbor, MI, USA), washed with pure ethanol (99.8%) and deionized water, and dried in the oven for 24 h at 70 °C. The resulting powders were then used for several physicochemical characterization techniques. 

#### 4.3.1. Phase Identification 

Chemical composition and structure were determined by FTIR (Fourier Transform Infrared Spectroscopy) (Bruker Vertex 70, Billerica, MA, USA), XRD (X-ray diffraction) (PANalytical Xpert PRO MRD, Malvern, United Kingdom), EDX based on different sections of the sample (Energy-dispersive X-ray spectroscopy) (FEI Quanta 400FEG ESEM/EDAX Genesis X4M, Hillsboro, EUA), and Atomic Absorption Spectrometry (λ = 422.7 nm, Flame: air-acetylene, with addition of lanthanum) (Perkin Elmer AAnalyst 400, Massachusetts, EUA) and UV-Vis Spectrophotometry-Molybdenum Blue (UV-Vis-Shimadzu UV-1800) to determine calcium and phosphate concentration. 

#### 4.3.2. Size, Morphology, and Crystallinity 

For SEM (scanning electron microscopy) (FEI Quanta 400FEG ESEM/EDAX Genesis X4M, Hillsboro, EUA with an accelerating voltage of 25.00 XkV and a working distance of 10 mm), analysis powders were covered by a 10-nm gold layer, and for TEM (transmission electron microscopy) (JEOL JEM-1400, Tokyo, Japan), suspensions of the obtained powders were prepared in ethanol and placed on a copper grid (StrataTek, Square Mesh Grids, 400 mesh, Copper, Ashburn, EUA). Finally, for particle size distribution analysis, suspensions were collected at the end of each experiment and directly analyzed by laser diffraction using ethanol as a solvent (LS 230, Beckman Coulter, Brea, CA, USA).

### 4.4. Cell Culture and Sample Preparation 

#### 4.4.1. HDFs Seeding 

Human Dermal Fibroblasts (HDFs) obtained from Innoprot (Ref: P10856-7-8) were grown as a monolayer under standard conditions (37 °C, 5% CO_2_) in basal medium consisting of Dulbecco’s Modified Eagle’s Medium (DMEM-Dulbecco’s Modified Eagle Medium; Gibco™ DMEM, high glucose), supplemented with 10% fetal bovine serum (FBS; BioWest), and 1% of penicillin-streptomycin solution (Lonza, Basel, Switzerland). At confluence, cells were detached from the culture flasks using TrypLE Express (Gibco, Waltham, MA USA), centrifuged, resuspended in culture medium, and seeded at a density of 1 × 10^4^ cells per cm^2^.

#### 4.4.2. Particle Preparation and Sterilization 

The particles obtained in the MOFPR were grinded in a ceramic mortar, in order to obtain a homogeneous powder. Subsequently, sterilization was achieved by using UV radiation for 15 min. Basal medium with 50 μg/mL of the particles produced in the MOFPR and commercial HAp (900203, Sigma Aldrich, Missouri, EUA) were prepared, according to previous works [12], which showed that this concentration is optimal for performing in vitro tests.

#### 4.4.3. Exposure of HDFs to Ca-Based Particles

After 24 h of attachment, cells were cultured under static conditions with the prepared particle suspensions. Cells were cultured for 1, 3, and 7 days in the presence of the produced particles.

### 4.5. Cell Interaction Evaluation

For DNA quantitation and 3-[4,5-dimethylthiazol-2-yl]-2,5-diphenyl tetrazolium bromide (MTT) assay, 48-well plates were used, with triplicates (*n* = 3, per time point). SEM and confocal tests were performed using non-adherent 24-well plates with tissue culture polystyrene coverslips (TCPS) (*n* = 2). For each time point, duplicates and one set of tests were included for SEM and confocal, respectively. The blank, negative, and positive controls consisted of DMEM, HDFs cultured in DMEM, and HDFs cultured in DMEM with 20% dimethyl sulfoxide (DMSO), respectively.

#### 4.5.1. MTT Staining

MTT assay was performed to evaluate the cell viability of the produced HAp particles. The seeded cells were treated with 0.5 mg/mL of MTT reagent (200 μL/well) for 2 h at 37 °C. DMSO reagent (200 μL) was added to dissolve the formazan crystals produced from the reduction of the tetrazolium salt. The absorbance of each sample was measured using a microplate reader at 570 nm (BioTek Synergy H1, Winooski, VT, USA). Cell viability was calculated according to the following equation:(1)$$Cell\;viability=\frac{Sample\;Mean\;Absorvance-Mean\;Blank}{Control\;Mean\;Absorvance-Mean\;Blank}\times 100$$

#### 4.5.2. DNA Quantification

To evaluate cell proliferation of the experimental conditions studied, DNA quantification was performed. At each time point, cells were washed twice with PBS and 1 mL of ultra-pure H_2_O was added to each well. Then, the cell suspension of each well was collected to an Eppendorf and stored at −80 °C until analysis. The amount of double stranded DNA (dsDNA), that is directly proportional to the cell number, was determined using the DNA quantitation assay (Quant-iT™ PicoGreen™ dsDNA Assay Kit, ThermoFisher, Waltham, MA, USA), according to the manufacturer’s instructions for multi-well plate assays [64]. Briefly, PicoGreen selectively binds dsDNA, which is measured fluorometrically at an excitation wavelength of 480 nm and an emission wavelength of 520 nm. A standard DNA concentration curve was plotted with standard amount of stock solution of dsDNA.

MTT and DNA data were analyzed as mean ± standard deviation with *n* = 9 for each bar. Statistical analysis was performed with GraphPad Prism 8.0 (GraphPad Software) using the two-way ANOVA test followed by Turkey’s method as multiple comparison post-test. The significance level was * *p* < 0.05, ** *p* < 0.01, *** *p* < 0.001, **** *p* < 0.0001.

#### 4.5.3. SEM and Confocal

Cell adhesion, proliferation, and morphology was assessed by SEM and Confocal. For SEM, at the end of each time-point, the cell-seeded constructs (TCPS seeded and cultured with HDFs and the different experimental conditions) were washed with PBS solution and fixed with a 2.5% (*v*/*v*) glutaraldehyde solution prepared in PBS. The solution acted during a period of 30 min at room temperature. After this fixation, a water wash was performed twice, and the samples were kept in the refrigerator until analysis. The day before SEM analysis, the samples were dehydrated using increasing concentrations of ethanol solutions (10%, 30%, 50%, 70%, 90%, 100% *v*/*v*) for 15 min for each concentration and air dried inside the flow chamber. In addition to using the secondary electron mode, the Z-contrast mode was also used for the experimental condition with cerium (CaP-SS-Ce).

Regarding confocal, at the end of 1 and 7 days of cell culture, the cell-seeded constructs were washed with PBS solution and fixed with a 500-µL formalin solution. After 30 min at room temperature, the wells were washed with water and left in sterile ultrapure water until observation. The day before observation, the cells were stained for filamentous actin (F-actin) and nuclei (DAPI). Briefly, samples were washed with PBS, fixed for 20 min in 4 wt.% paraformaldehyde (Sigma), and permeabilized with 0.2% Triton X-100 (Sigma) for 7 min. After washing with PBS, Phalloidin-TRITC (Sigma) diluted at 1:500 was incubated for 1 h. After washing with PBS, DAPI (Sigma) diluted at 1:1000 was incubated and further washed with PBS. The stained samples were further observed under laser scanning confocal microscopy–Zeiss LSM900 confocal microscope (Zeiss, Jena, Germany). The scanned Z-series were projected onto a single plane and pseudo-colored using ImageJ.

### 4.6. Antibacterial Activity

A preliminary test to evaluate the antibacterial potential of the synthesized particles was conducted using CaP-SS-Ce. The particles were incubated for 24 h with Gram+ (*Staphylococcus aureus*) and Gram– (*Escherichia coli*) bacteria, using the same concentration used for cell culture (50 µg/mL) and a higher concentration (12,500 µg/mL). 

## 5. Conclusions

The CaPs herein produced, and their modified formulations, can be safely used for direct application in skin wound healing, due to the crucial role of calcium in promoting homeostasis in skin regeneration. We have demonstrated that HAp-based nanoparticles with rod/plate shape (CaP1-nano, CaP2-nano/micro, and CaP-SS) promoted fibroblast adhesion and proliferation. The coprecipitation of sericin with CaP functionalized these “old” bioceramics, with a strong impact on cell proliferation. The natural bioactive protein sericin is known to shape CaP mineralization, while inducing a favorable cell metabolic activity towards regeneration. The antibacterial properties of the materials were successfully achieved for *Escherichia coli*, by adding cerium to the system, without compromising the biocompatibility.

Our fabrication strategy, using a MOFPR, allowed the adequate mixing in a multiphase system, generating highly homogeneous nanoparticles. The possibility of synthetizing tailored particles by changing the oscillation mixture (*f* and *x*_0_) and experimental conditions (initial concentrations) is also of high importance in the biomedical field. Furthermore, its continuous mode allows the process scale-up, with no impact on the final physicochemical properties of the materials, which makes its use of great interest to the biomedical industry.

## Figures and Tables

**Figure 1 molecules-27-02249-f001:**
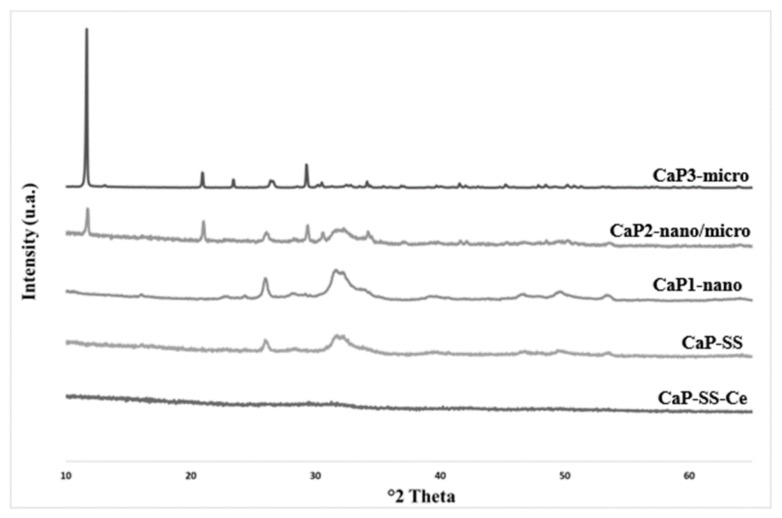
XRD patterns of the experimental conditions analyzed.

**Figure 2 molecules-27-02249-f002:**
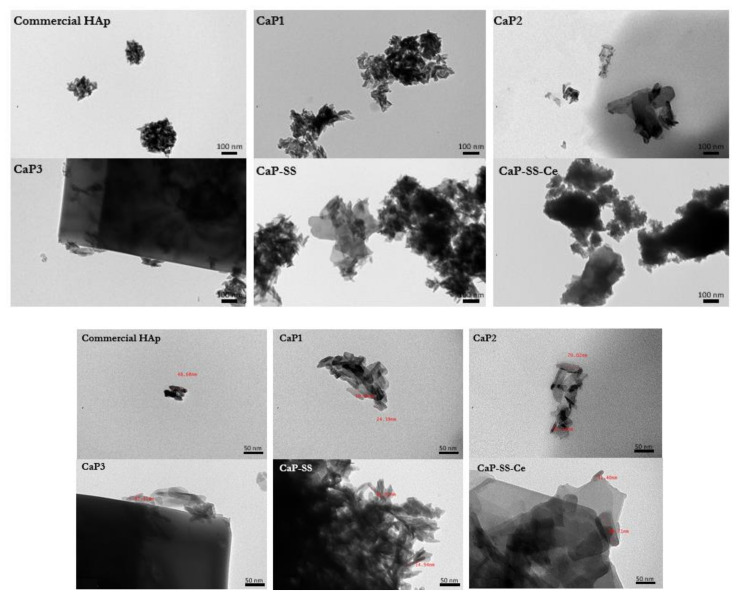
TEM images of the particles produced.

**Figure 3 molecules-27-02249-f003:**
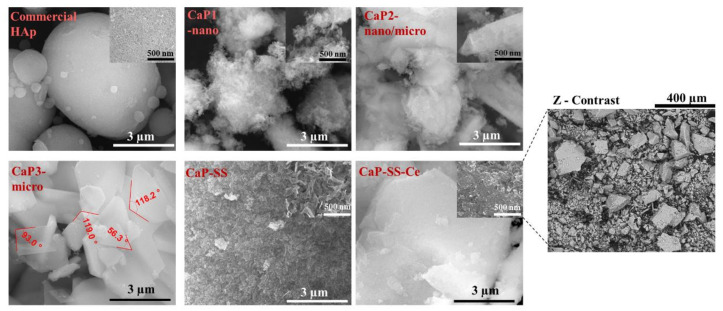
SEM images of the produced particles.

**Figure 4 molecules-27-02249-f004:**
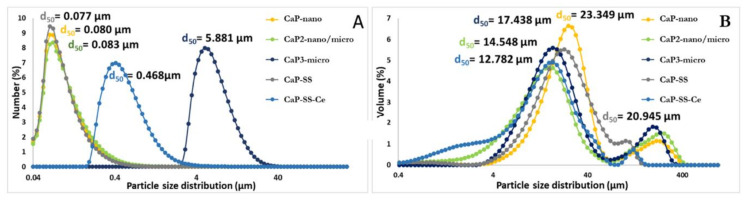
Particle size distribution in (**A**) number and (**B**) volume.

**Figure 5 molecules-27-02249-f005:**
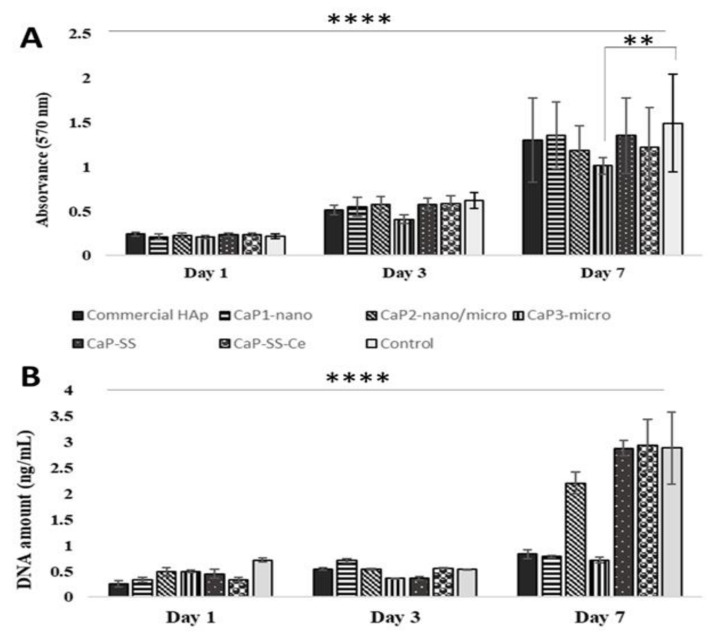
Cell viability and proliferation measured by (**A**) MTT assay and (**B**) DNA quantification, for HDFs cultured in direct contact with synthesized and commercial HAp particles, for 1, 3, and 7 days of culture (** *p* < 0.01, **** *p* < 0.0001).

**Figure 6 molecules-27-02249-f006:**
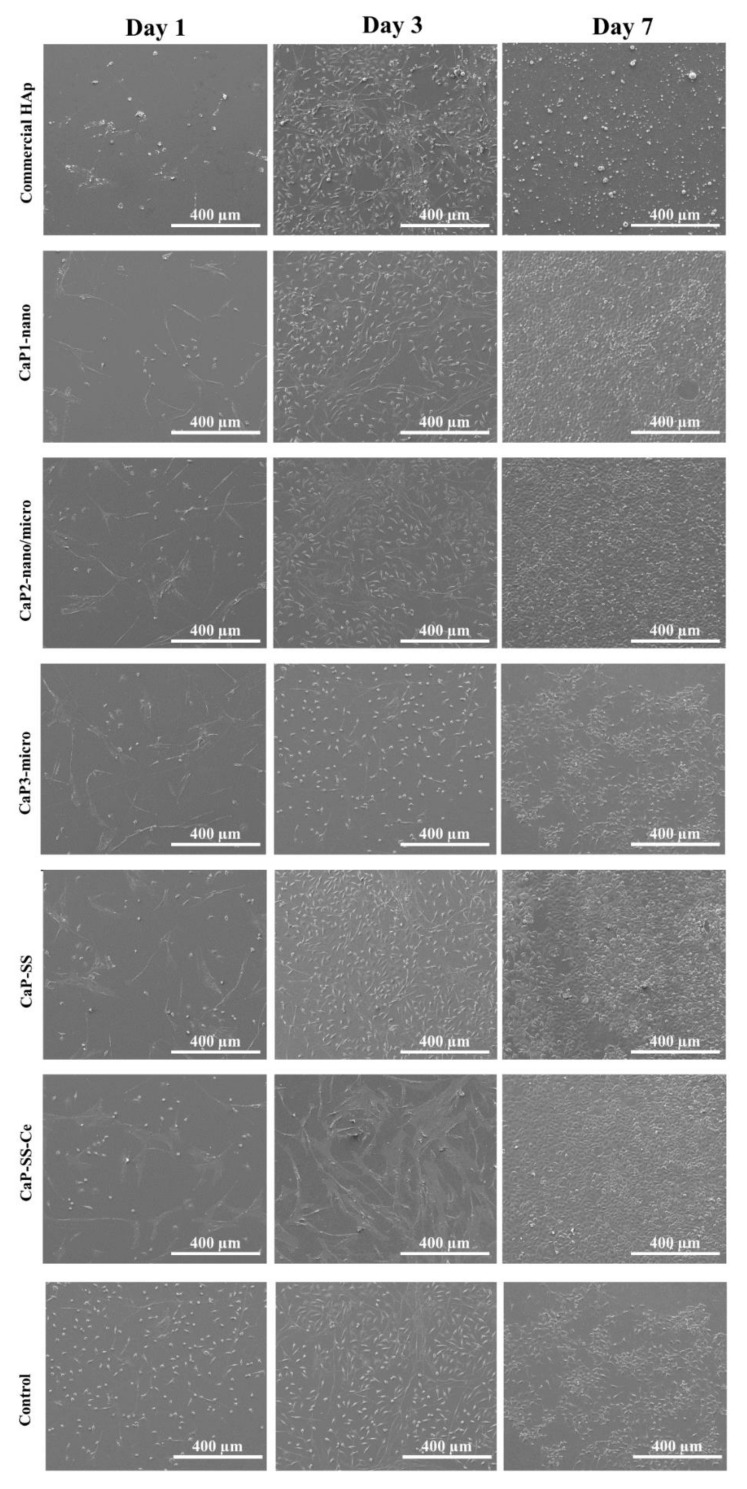
SEM images illustrating the morphology, distribution, and proliferation of HDFs cultured in direct contact with the produced Ca-based particles and commercially available HAp for 1, 3, and 7 days of culture.

**Figure 7 molecules-27-02249-f007:**
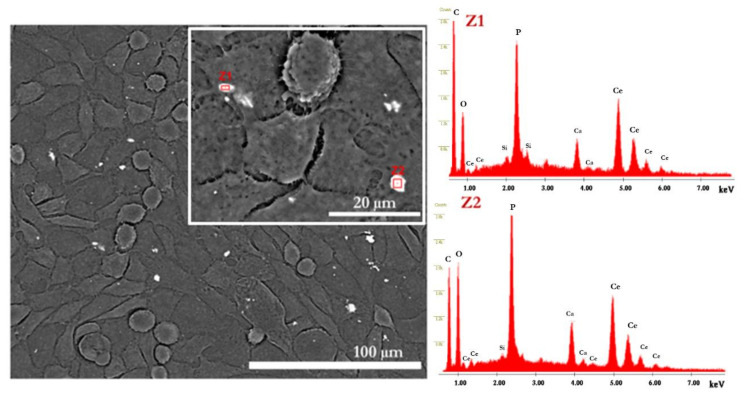
SEM images using the Z-contrast mode for CaP-SS-Ce particles and EDX spectra after 7 days of culture.

**Figure 8 molecules-27-02249-f008:**
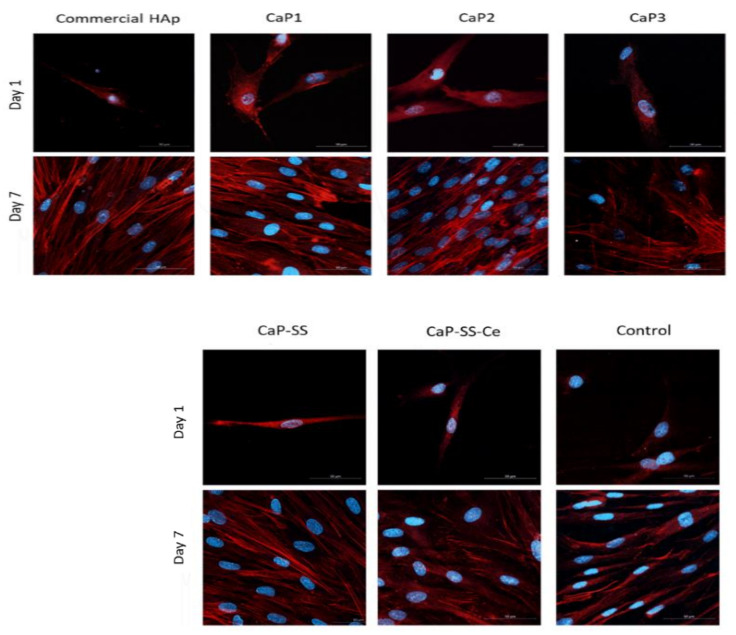
Confocal images illustrating the morphology, distribution, and proliferation of the HDFs’ nucleus and cytoskeleton cultured in direct contact with the produced Ca-based particles and commercially available HAp for 1 and 7 days of culture.

**Figure 9 molecules-27-02249-f009:**
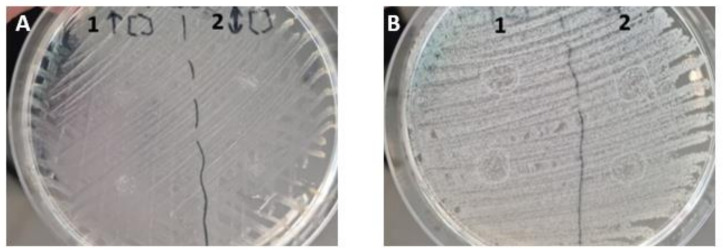
Preliminary antibacterial test on CaP-SS-Ce particles using (**A**) Gram+ (*Staphylococcus aureus*) (**B**) Gram– (*Escherichia coli*) bacteria and 20 µL of a particle suspension in Ringer’s solution with (1) 50 µg/mL, (2) 12,500 µg/mL.

**Figure 10 molecules-27-02249-f010:**
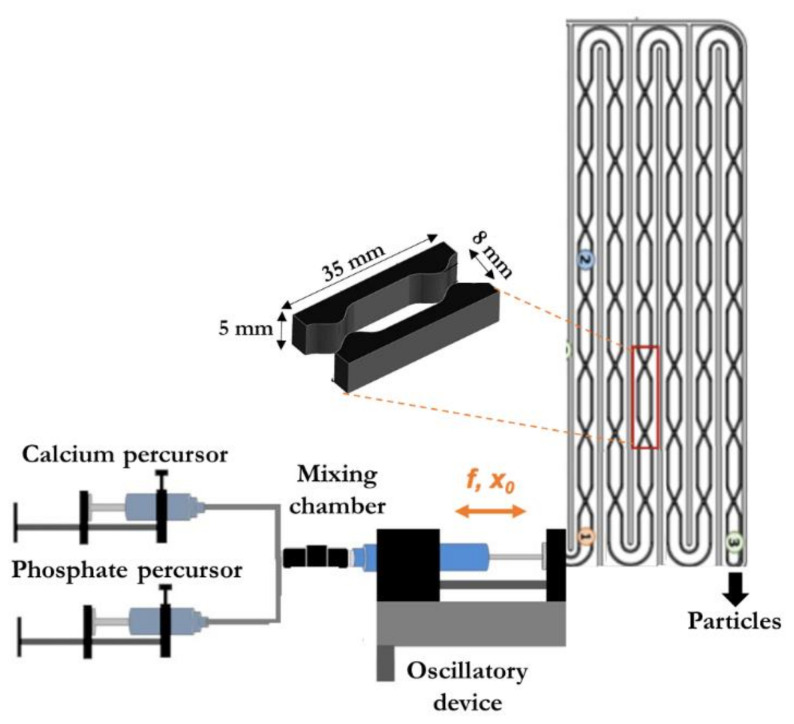
Experimental set-up adapted from [62].

**Table 1 molecules-27-02249-t001:** Identification of the functional groups present in the FTIR spectra of the samples synthesized.

Wavenumber (cm^−1^) Functional Groups	HAp Reference	Commercial HAp	Sericin	CaP1-Nano	CaP2-Nano/Micro	CaP3-Micro	CaP-SS	CaP-SS-Ce
**PO_4_^3−^**	1087, *ν3*	1090	-	-	-	-	-	-
	1032, *ν3*	1024		1020	1020		1020	1023
	962, *ν1*	962		961	960		961	-
	602, *ν4*	600		600	597		600	600
	561, *ν4*	561		558	561		559	530
	472, *ν2*	-		-	-		-	-
**HPO_4_^3−^**	-	-	-	-	-	983	-	-
						1053		
**P-O**	-	-	-	-	-	574	-	-
						541		
**CO_3_^2−^**	875	874	-	866	875	869	865	-
	1410	1418		-	-	-		
**OH^−^**	631	630	-	-	-	-	-	-
	3572	3571		-	-	-	-	-
**ⱱO-H**	-	-	-	-	-	3533		
						3477		
						3151		
**H-O-H**	-	-	-	-	-	1647	-	-
**P=O**	-	-	-	-	-	1205		
						1099		
**P-O-P**						777		
						653		
**Amide I**	-	-	1657	-	-	-	1644	1644
**Amide II**	-	-	1551	-	-	-	-	-
**Amide III**	-	-	1251	-	-	-	-	-

**Table 2 molecules-27-02249-t002:** Identification of the peaks present in the XRD patterns of the synthetized samples and reference CaPs.

Sample	°2 Theta								
Miller Index	HAp Reference (JCPDS 00-009-0432)	Brushite (JCPDS 72-0713	Commercial HAp	[17]	CaP1-Nano	CaP2-Nano/Micro	CaP3-Micro	CaP-SS	CaP-SS-Ce
020	-	11.71	-			11.71	11.70	-	-
121		20.62				21.00	21.04		
104		23.47				-	23.53		
002	25.79	-	26.09	25.90	25.95	26.07	-	25.96	
141	-	29.15	-			29.33	29.38		
211	31.08	-	32.08	31.86	31.78	31.64	-	31.77	
112	32.20		32.41	32.20	32.26	32.29		32.27	
300	32.92		33.17	32.90	33.73	-		33.79	
121	-	34.19	-			34.18	34.21		
222		36.49				-	36.57		
310	39.81	-	40.12	39.86	39.88	39.78	-	40.59	
152	-	41.01	-			41.67	41.69		
222	46.69	-	46.95	46.69	46.67	46.61	-	46.81	
213	49.43		49.74	48.16	49.82	50.31		49.60	
143	-	50.75	-				50.25	-	
004	53.21	-	53.43	53.27	53.37	53.56	-	53.51	

**Table 3 molecules-27-02249-t003:** EDX analysis to the CaP and CaP composites produced and Ca/P molar ratio.

Calcium-Based Material	Oxygen (O)		Phosphate (P)		Calcium (Ca)		Cerium (Ce)		Ca/P Molar Ratio
	Weight (%)	Atomic (%)	Weight (%)	Atomic (%)	Weight (%)	Atomic (%)	Weight (%)	Atomic (%)	
Commercial HAp	40.26	60.68	19.09	14.86	40.64	24.45			-
CaP1-nano	40.76	60.97	20.87	16.10	38.38	22.92			1.68
CaP2-nano/micro	35.19	55.24	22.45	18.20	42.37	26.44			1.69
CaP3-micro	45.75	65.32	22.37	16.50	31.88	18.17			1.23
CaP-SS	41.30	61.42	21.42	16.44	37.28	22.15			1.65
CaP-SS-Ce	30.38	58.78	18.67	18.65	20.00	15.44	30.51	6.74	1.01

**Table 4 molecules-27-02249-t004:** Cell viability after 7 days of cell culture (Equation (1)).

Experimental Conditions	Cell Viability (%)
Commercial HAp	87.4 ± 27.5
CaP1-nano	90.9 ± 21.26
CaP2-nano/micro	79.4 ± 18.88
CaP3-micro	67.7 ± 2.26
CaP-SS	90.5 ± 28.74
CaP-SS-Ce	96.8 ± 24.56

**Table 5 molecules-27-02249-t005:** Significance level for DNA quantification after 7 days of cell culture (** *p* < 0.01, *** *p* < 0.001, **** *p* < 0.0001, ns-not significant).

	CaP1-Nano	CaP2-Nano/Micro	CaP3-Micro	CaP-SS	CaP-SS-Ce	Control
Commercial HAp	ns	**	ns	****	****	****
CaP1-nano		***	ns	****	****	****
CaP2-nano/micro			***	ns	ns	ns
CaP3-micro				****	****	****
CaP-SS					ns	ns
CaP-SS-Ce						ns

**Table 6 molecules-27-02249-t006:** CaPs and CaP/sericin-based composites produced in the MOFPR. Silk sericin and cerium concentrations used were based on published works [3,31].

Experimental Conditions	Initial Reagents Concentration	Description	Physicochemical Characteristics	Frequency (Hz)	Amplitude (mm)	[Sericin] g/L	[Cerium] g/L
1.	CaCl_2_.2H_2_O (0.02 M)	CaP1-nano	HAp nanoparticles	1.9	4	-	-
2.	Na_2_HPO_4_ (0.012 M)	CaP2-nano/micro	HAp nano and microparticles	4	4		
3.	CaCl_2_.2H_2_O (0.2 M)	CaP3-micro	Brushite particles	1.9	4		
	Na_2_HPO_4_ (0.12 M)						
4.	CaCl_2_.2H_2_O (0.02 M)	CaP-SS	To evaluate			0.1	
5.	Na_2_HPO_4_ (0.012 M)	CaP-SS-Ce				0.1	0.4

## Data Availability

The data presented in this study are available on request from the corresponding author. The data are not publicly available since all graphical results of the processed data and statistical analyses are included in this paper.

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
