# Peer review of "Continuous Production of Highly Tuned Silk/Calcium-Based Composites: Exploring New Pathways for Skin Regeneration"

_molecules, 2022, doi:10.3390/molecules27072249_

Round 1
Reviewer 1 Report
Comments:
p.3,l 100:“... and 1647 - 653 cm-1, result from νO–H vibration modes and δO-H bend modes of H2O in brushite…“ This is a very general statement, as the PO4 and CO3 vibration peaks also fall into this region.Please, correct this sentence.
p.3, XRD: How was crystallinity of precipitated calcium phosphates?
p.4: How was standard deviations of means? EDX analysis is a semi-quantitative method, which also requires compliance with standard analysis conditions (e.g. flat surface without pores) and is not suitable for accurate determination of element contents in powders. I strongly recommend to add chemical analysis of powders including determination of Ca/P ratio or Ce content. From text it is not clear how were real composition of powders. In addition, it is not correct to show results of EDX analysis with three decimals due to errors. Besides samples were coated with gold which strongly overlapped with phosphorus peak in EDX spectrum which is another source for incorrect determination of phosphorus content.
p.7:“...At the nanoscale it is possible to verify that these microstructures are composed…“ It is improper expression -they are not microstructures, but the microstructure of nanoparticle agglomerates.
p.7: The whole evaluation of the morphology and distribution of particles is insufficient - separately the morphology of agglomerates (SEM), the morphology of nanoparticles in agglomerates (TEM) should be assessed, and the overall distribution given on the curves should be discussed - why they differ e.g. is there an obvious contradiction that CaPnano has a high volume fraction of 4-40 μm particles on curve B, but they have no representation on curve A. In addition, it would be appropriate to do SAED particle analysis by TEM.
Fig.5: The statistical analysis in the figure (also in table) is very confusing. I believe that to compare the statistical difference between 1 or 3 days and 7 days was unnecessary because it is absolutely clear difference between these means.
p.10, l 187:“...it allows to clearly demonstrate the interaction of the particles with the HFD’s.“What did authors mean-how interaction is visible in figure, please decribe this…
p.12, l293: „...achieved in SEM Z -contrast mode due to the atomic structure composition of the cerium, demonstrates that this element is well distributed across…“ Z mode do not characterize distribution of Ce but EDX mapping or line scan is much better alternative for correct determination of cerium distribution in particles especially by STEM (due to better resolution at higher magnification). I recommend to be very carefull with such a evaluation in manuscript.
l455:How was the pH of the solutions during precipitation?How was the final real amount of sericin in the composites?Please add these characteristics to text.
Author Response
We would like to thank the reviewer comments on the manuscript " Continuous Production of Highly Tuned Silk/Calcium-Based Composites: Exploring New Pathways for Skin Regeneration" for the Special Issue "Nanotechnology: Engineering the Future of Medicine" of Molecules (ID: molecules-1611644). The manuscript has been modified and improved according to the suggestions of the reviewer. The responses to the reviewer comments can be found in the word file below.

Reviewer 2 Report
In this manuscript, Veiga et al reported a method to synthesize calcium phosphate-based nanoparticles and composites. The biological behavior of as-prepared materials is evaluated using human dermal fibroblasts, and antibacterial properties of Ce-dosed particles were characterized. The results are interesting for tissue engineering, and the data supported the conclusions. However, some questions need to be addressed before the acceptance of this manuscript.
- Stability of the materials are important for biological applications. Therefore, size of the particles should be characterized over a long period of time (eg. several weeks) to show the stability of the materials.
- Please double-check that Figure 5A is correct. From Figure 5A, viability of the cells is decreasing instead of increasing over time. Also, data from the control group are missing. It is also suggested that the author use different colors to label different groups. In current settings, the colors are difficult to differentiate from each other.
- In Figure 8, the control group is missing.
- In the manuscript, the authors claim that there is an increase of cell density for all groups. For the CaP-based materials, is this effect concentration-dependent? Cells should be treated with different concentrations of materials to show this effect.
- Figure 9 shows antibacterial behavior of CaP-SS-Ce samples. What about other materials? Do they still have similar properties even if they don’t have Ce?
Author Response
We would like to thank the reviewer comments on the manuscript " Continuous Production of Highly Tuned Silk/Calcium-Based Composites: Exploring New Pathways for Skin Regeneration" for the Special Issue "Nanotechnology: Engineering the Future of Medicine" of Molecules (ID: molecules-1611644). The manuscript has been modified and improved according to the suggestions of the reviewer.
Please see the attachment with the responses to the reviewer comments.

Reviewer 3 Report
- line 99: “The peaks identified in Table 2 between 3533 and 3151 cm-1 and 1647 - 653 cm-1, result from” – please, replace Table 2 with Table 1.
- Fig. 2 – Black and white text is poorly visible on SEM pictures. Please, use a color.
- Please, provide both the trade name and the producer of Commercial HAp.
- Fig. 6 – All images are of a poor quality. Almost nothing is visible.
- line 218: “brushite (also known as dibasic calcium phosphate dehydrate).” – please, replace dehydrate by dihydrate (this commonly happens due to an automatic autocorrection option of MS Word).
Author Response
We would like to thank the reviewer comments on the manuscript " Continuous Production of Highly Tuned Silk/Calcium-Based Composites: Exploring New Pathways for Skin Regeneration" for the Special Issue "Nanotechnology: Engineering the Future of Medicine" of Molecules (ID: molecules-1611644). The manuscript has been modified and improved according to the suggestions of the reviewer.
Please see the attachment (word file) with the responses to the reviewer comments.

Round 2
Reviewer 1 Report
The crystallinity could analysed using the Scherrers eq. not only vague statment.
Reviewer 2 Report
The authors have addressed my questions after the revision, and the paper can be accepted after some minor text editing and proof reading.